# Castration Induces Down-Regulation of A-Type K^+^ Channel in Rat Vas Deferens Smooth Muscle

**DOI:** 10.3390/ijms20174073

**Published:** 2019-08-21

**Authors:** Susumu Ohya, Katsunori Ito, Noriyuki Hatano, Akitoshi Ohno, Katsuhiko Muraki, Yuji Imaizumi

**Affiliations:** 1Department of Pharmacology, Graduate School of Medical Sciences, Nagoya City University, Nagoya 467-8601, Japan; 2Department of Molecular and Cellular Pharmacology, Graduate School of Pharmacological Sciences, Nagoya City University, Nagoya 467-8603, Japan; 3Laboratory of Cellular Pharmacology, Aichi Gakuin University, Nagoya 464-8650, Japan

**Keywords:** A-type K^+^ channel, K_V_4.3, castration, testosterone, vas deferens, smooth muscle

## Abstract

A-type K^+^ channels contribute to regulating the propagation and frequency of action potentials in smooth muscle cells (SMCs). The present study (i) identified the molecular components of A-type K^+^ channels in rat vas deferens SMs (VDSMs) and (ii) showed the long-term, genomic effects of testosterone on their expression in VDSMs. Transcripts of the A-type K^+^ channel α subunit, Kv4.3L and its regulatory β subunits, KChIP3, NCS1, and DPP6-S were predominantly expressed in rat VDSMs over the other related subtypes (Kv4.2, KChIP1, KChIP2, KChIP4, and DPP10). A-type K^+^ current (I_A_) density in VDSM cells (VDSMCs) was decreased by castration without changes in I_A_ kinetics, and decreased I_A_ density was compensated for by an oral treatment with 17α-methyltestosterone (MET). Correspondingly, in the VDSMs of castrated rats, Kv4.3L and KChIP3 were down-regulated at both the transcript and protein expression levels. Changes in Kv4.3L and KChIP3 expression levels were compensated for by the treatment with MET. These results suggest that testosterone level changes in testosterone disorders and growth processes control the functional expression of A-type K^+^ channels in VDSMCs.

## 1. Introduction

Testosterone, the male sex hormone is responsible for the virilization of the Wolffian duct system into the epididymis, vas deferens, and seminal vesicle [1]. The testosterone-mediated development of the Wolffian duct is regulated by a number of growth factors, including epidermal growth factor (EGF), insulin-like growth factor (IGF), and fibroblast growth factor (FGF) [2]. Castration significantly decreases serum testosterone levels, and this is accompanied by significant reductions in the weight of the vas deferens (VD) [3]. Previous studies reported that spontaneous contractions appeared in the VD of castrated animals, and testosterone replacement suppressed these phenomenon [4,5,6]. Boselli et al. (1994) indicated that castration decreased K^+^ conductance in VD smooth muscle cells (VDSMCs) [6]. VDSMCs functionally express rapidly-inactivating, voltage-dependent (A-type) K^+^ channels [7]. However, the long-term, genomic effects of testosterone on the gene expression of A-type K^+^ channel subunits remain to be elucidated in VDSMs.

Kv4 channels produce rapidly-inactivating K^+^ currents (I_A_) and regulate propagation and frequency of action potentials in SMCs with relatively high excitability, such as urinary bladder and vas deferens [8]. Molecular biological approaches revealed that Kv4 pore-forming α subunits assembled with auxiliary β subunits to create A-type K^+^ channel complexes with unique attributes and functions, and the following Kv4β subunits were identified—K^+^ channel-interacting proteins (KChIP1-4), K^+^ channel-activating protein (KChAP), neuronal Ca^2+^ sensor 1 (NCS1), and dipeptidyl peptidase-like proteins (DPP6 and 10) [9]. Kv4β subunits modulate the activity and kinetics of the Kv4α subunit, and also control membrane trafficking [10,11,12,13]. Therefore, the functional expression of A-type channels is not only controlled by the levels of Kv4α subunits, but also requires their association with Kv4β subunits. Several research groups have reported that the expression of Kv4α subunits is associated with pathophysiological conditions such as heart failure and diabetes [14,15,16]. During pregnancy, the expression of the Kv4α and Kv4β subunits was found to be dynamically and region-selectively regulated in the myometrium [17,18]. We cloned and characterized the longer variant of Kv4.3 (Kv4.3L) in rat VDSMs using molecular biology and patch-clamp electrophysiology [19]; however, the mechanisms underlying the testosterone-mediated regulation of Kv4α and β subunit expression in VDSMs currently remain undetermined.

In the present study, the molecular components of Kv4β subunits were identified in the rat VDSMs, and the expression levels of Kv4α and Kv4β subunits in rat VD were compared between the following rats using a real-time PCR analysis, Western blots, and electrophysiological approaches—(i) sham-operated, (ii) castrated, (iii) castrated with a 17α-methyltestosterone (MET; a synthetic derivative of testosterone given by the oral route without loss of bioavailability) treatment, and (iv) castrated with vehicle treatment. The results obtained suggest that the testosterone level changes in male hypogonadism and certain growth processes may control A-type K^+^ channel expression in VDSMs.

## 2. Results

### 2.1. Molecular Identification of Kv4 Channel α and β Subunits Expressed in the Rat VD

Rats were divided into four experimental groups—(i) sham-operated, (ii) castrated, (iii) MET-treated, and (iv) vehicle-treated (see Section 4.1.). Similar to the previous study by Wakade et al. (1975) [20], histological examinations of the four groups indicated that the wet weights of the VD, seminal vesicles, and the prostate were significantly reduced by almost 80%, 4 weeks after castration (*n* = 9) (Figure 1). The administration of MET almost completely reversed these changes (Figure 1).

We measured the gene expression levels for the Kv4α and Kv4β subunits in the VD of sham-operated rats using quantitative real-time PCR. Gene-specific PCR primers were designated for Kv4.2, Kv4.3L, KChIP1–4, KChAP, NCS1, DPP6, and DPP10. Data were expressed as relative to the mRNA levels of β-actin (ACTB). In the rat VD, Kv4.3L, KChIP3, NCS1, and DPP6 were predominantly expressed, and the expression relative to ACTB was 0.051 ± 0.005, 0.021 ± 0.001, 0.020 ± 0.001, and 0.066 ± 0.006, respectively (*n* = 6 for each, Figure 2). On the other hand, the expression of Kv4.2, KChIP1, KChIP2, KChIP4, KChAP, and DPP10 was less than 0.01 (*n* = 6). These results suggest that the main components of I_A_ in rat VDSM are Kv4.3L, KChIP3, NCS1, and DPP6.

### 2.2. Changes in A-Type K^+^ Current Properties in Rat VDSMCs by Castration

Figure 3A showed that transient outward currents (I_A_) were elicited by the 1000 ms depolarizing-voltage-step between −70 and +40 mV from a holding potential (−80 mV) with 10 mV increments, once every 10 s in rat VDSMCs—(a) sham-operated, (b) castrated, (c) MET-treated, and (d) vehicle-treated. In order to isolate I_A_, L-type Ca^2+^ currents, delayed rectifier K^+^ currents, and Ca^2+^-activated K^+^ currents were blocked by addition of 1.2 mM CdCl_2_ and 30 mM tetraetylammonium chloride (TEA) in the external medium and 5 mM ethylene glycol-bis(2-aminoethylether)-N,N,N’,N’-tetraacetic acid (EGTA) in the pipette solution (see Section 4.4.). The membrane capacitance of VDSMCs in the sham-operated, castrated, MET-treated, and vehicle-treated groups were 53.2 ± 2.5, 39.8 ± 0.9, 53.2 ± 2.3, and 34.8 ± 1.4 pF, respectively (*n* = 15–20), and were significantly smaller in the castrated and vehicle-treated groups than in the sham-operated and MET-treated groups (*p* < 0.01). The current density–voltage relationship of peak outward current was shown in Figure 3B. In VDSMCs of normal rats, I_A_ density was not affected by MET treatment (not shown). In each group, the threshold of I_A_ was the same as approximately −30 mV. The peak I_A_ density at +40 mV was approximately 30% lower in the castrated and vehicle-treated groups than in the sham-operated and MET-treated groups—23.2 ± 0.9 (sham-operated, *n* = 15), 15.8 ± 0.9 (castrated, *n* = 20), 20.7 ± 0.8 (MET-treated, *n* = 17), and 16.4 ± 1.3 (vehicle-treated, *n* = 16) pA/pF, respectively (Figure 3C, Table 1). The times to the peak current at +20 mV in these groups were 4.35 ± 0.25, 4.13 ± 0.23, 4.65 ± 0.25, and 4.44 ± 0.26 ms, respectively, showing no significant differences between the four groups (*p* > 0.05) (Table 1). Furthermore, no significant differences were observed at different potentials of 0, +10, +30, and +40 mV (not shown). The times to half-inactivation from the peak at +20 mV in four groups were 18.0 ± 1.8, 14.6 ± 0.6, 19.7 ± 1.4, and 19.3 ± 1.84 ms, respectively, showing no significant differences between the groups (*p* > 0.05) (Table 1). In the four groups, the inactivation time course of I_A_ was fit well by one exponential component, and more than 95% of inactivation was due to fast inactivation, with no significant differences being observed in τ values.

The voltage dependence of activation and steady-state inactivation was assessed using the conventional double pulse protocol, as described in Section 4.5. Half-activation voltage (V_1/2_) and slope factors (S) were obtained by fitting the Boltzmann equation to each set of data, as summarized in Table 1. In the sham-operated, castrated, MET-treated, and vehicle-treated groups, V_1/2_ were −13.9 ± 2.4, −9.2 ± 3.5, −10.4 ± 6.1, and −11.3 ± 4.0 mV, and S were 12.7 ± 0.7, 9.0 ± 1.0, 11.7 ± 1.0, and 13.5 ± 1.8, respectively (*n* = 5–6), and no significant differences were found. The half-inactivation voltage (V_1/2_) and S were also obtained by fitting the Boltzmann equation (Figure 4A). V_1/2_ were −58.6 ± 1.4, −57.0 ± 1.5, −60.3 ± 1.1, and −57.7 ± 0.9 mV, and S were 4.5 ± 0.2, 5.0 ± 0.2, 4.6 ± 0.1, and 5.4 ± 0.2, respectively (*n* = 14–16), and no significant differences were found (Table 1). The time course of recovery from inactivation was measured by applying a paired pulse protocol from a holding potential of −80 mV. Recovery from inactivation was defined as the relative amplitude of the peak current elicited by the second pulse versus that by the first one and plotted as a function of the interval between paired pulses. The time course was fitted by a single exponential function (Figure 4B). In the four groups, time constants (t) at −80 mV of the interpulse potential were 47.9 ± 8.9, 37.5 ± 3.1, 51.6 ± 5.2, and 50.2 ± 5.2 ms, and no significant differences were found between the four groups (Table 1).

### 2.3. Changes in the mRNA Expression of Kv4 Channel α and β Subunits Expressed in the Rat VD by Castration

To clarify whether testosterone regulates the gene expression of the Kv4α and Kv4β subunits in VDSMs, the transcriptional expression levels of Kv4.3L, KChIP3, NCS1, and DPP6 were compared in the four groups using a real-time PCR analysis (Figure 5). Castration produced an approximately 40% decrease in the mRNA level of Kv4.3L in VDSMs, and expression relative to ACTB was 0.051 ± 0.005 and 0.029 ± 0.001 in the sham-operated and castrated groups, respectively (*n* = 6 for each, *p* < 0.01 versus sham-operated). In addition, the MET treatment recovered the effects of castration, and expression relative to ACTB was 0.040 ± 0.001 (*n* = 6, *p* < 0.05 versus sham-operated; *p* < 0.01 versus castrated). In the vehicle-treated group, expression was similar to that in the castrated group (0.028 ± 0.003, *n* = 6, *p* < 0.01 versus the sham-operated and MET-treated) (Figure 5A). Castration produced an approximately 80% decrease in the mRNA levels for the Kv4β subunit, KChIP3 in the VD, and the expression relative to ACTB was 0.021 ± 0.001 and 0.005 ± 0.001 in the sham-operated and castrated groups, respectively (*n* = 6 for each, *p* < 0.01 versus the sham-operated). The MET treatment prevented the effects of castration, and expression relative to ACTB was 0.016 ± 0.002 (*n* = 6, *p* < 0.01 versus castrated). In the vehicle-treated group, expression was similar to that in the castrated group (0.005 ± 0.001, *n* = 6, *p* < 0.01 versus the sham-operated and MET-treated) (Figure 5B). No significant differences were observed in NCS1 or DPP6 expression between the four groups. The expression of NCS1 relative to ACTB was 0.020 ± 0.001, 0.017 ± 0.001, 0.019 ± 0.001, and 0.018 ± 0.001 (Figure 5C), while that of DPP6 was 0.066 ± 0.006, 0.091 ± 0.013, 0.058 ± 0.008, and 0.089 ± 0.007 (Figure 5D) in the sham-operated, castrated, MET-treated, and vehicle-treated groups, respectively. Castration did not induce the up-regulation of the other Kv4α and Kv4β subunits expressed at very low levels in normal rats (less than 0.005 relative to ACTB). These results suggest that the stoichiometry of Kv4β subunits in the VDSMs of castrated rats might differ from that in the VDSMs of normal rats.

Previous studies have suggested that (i) Kv7 channel β subunits (KCNE1-3) and the Na^+^ channel β_2_ subunit (Navβ_2_) modulate Kv4 channel properties in cardiac muscles and neurons [21,22], and (ii) significant decreases in serum testosterone levels might cause a switch in A-type K^+^ channel genes from Kv4.2 to Kv1.4 in cardiac muscles [3,15]. We compared the expression levels of KCNE1-3, Navβ_2_, and Kv1.4 in the four groups and found that their expression relative to ACTB was markedly less than 0.01 in the VDSMs of the four groups, with no significant changes being detected.

### 2.4. Changes in the Protein Expression of Kv4 Channel α and β Subunits in Rat VD by Castration

The expression levels of the Kv4.3L, KChIP3, NCS1, and DPP6 proteins in the VD were verified by Western blot analyses. As shown in Figure 6, the anti-Kv4.3 antibody recognized a band at approximately 65 kDa in VD membranes. A densitometric analysis revealed that castration produced an approximately 35% decrease (66.5 ± 7.2% of sham-operated, *n* = 4, *p* < 0.05), and the MET treatment prevented the effects of castration (107.7 ± 8.8% of sham-operated, *n* = 4, *p* < 0.05 versus castrated). Expression was similar between the vehicle-treated and castrated groups (65.2 ± 12.4%, *n* = 4, *p* < 0.05 versus the sham-operated and the MET-treated groups) (Figure 6A). The anti-KChIP3 antibody recognized a band at approximately 30 kDa in VD membranes. The densitometric analysis revealed that castration produced an approximately 85% decrease (15.1 ± 3.7% of sham-operated, *n* = 4, *p* < 0.01), while the MET treatment prevented the effects of castration (117.3 ± 10.0% of sham-operated, *n* = 4, *p* < 0.01 versus castrated). In the vehicle-treated group, expression was similar to that in the castrated group (20.4 ± 5.5%, *n* = 4, *p* < 0.01 versus sham-operated and MET-treated) (Figure 6B). No significant differences were observed in NCS1 (22 kDa) or DPP6 (100 kDa) expression between the four groups. In the castrated, MET-treated, and vehicle-treated groups, the expression of NCS1 was 102.2 ± 9.3, 102.6 ± 8.4, and 105.9 ± 9.2%, while that of DPP6 was 80.2 ± 5.2, 100.8 ± 15.9, and 102.6 ± 12.2% of the sham-operated group, respectively (Figure 6C,D). These results were consistent with those obtained on the expression levels of the Kv4α and Kv4β subunits in real-time PCR.

### 2.5. Identification of DPP6 Isoforms Expressed in the Rat VD

The results of the Western blotting analysis suggested the presence of an expression of the longer DPP6 isoform in the VDSM membrane of castrated rats. The DPP6 isoforms, DPP6-L and DPP6-S with different N-terminal cytoplasmic domains have been cloned from the rat brain [23,24]. Therefore, to identify the DPP6 isoform expressed in the VDSMs of the sham-operated and castrated rats, specific PCR primers for DPP6-L and DPP6-S were designated for the real-time PCR analysis. In contrast to our expectations, the DPP6-S transcript alone was expressed in the VDSMs of both groups at similar expression levels (Figure 7A). We also designed 8 pairs of PCR primers that over-lapped whole areas of DPP6-S, however, no spliced isoforms of DPP6-S were obtained in either groups (Figure 7B). Kin et al. (2001) showed that DPP6 is a mannoprotein with high mannose-type oligosaccharides in a heterologous expression system [23]. Glycosylation and mannosylation play crucial roles in intracellular trafficking and cell surface expression in membrane proteins, and DPP6 has its consensus sites on an extracellular domains [23]. The apparent molecular mass of DPP6 proteins in the sham-operated group was similar to the lower band of the enzymatically deglycosylated DPP6 protein in the castrated group (Figure 7C), suggesting that DPP6 proteins were deglycosylated in the VDSMs of sham-operated rats.

## 3. Discussion

A-type K^+^ channels contribute to regulating the propagation and frequency of action potential in VDSMs. In the androgen-dependent organ, VD, testosterone and its derivatives might control the A-type K^+^ channel activity, and this is accompanied by changes in the gene expression of its molecular components. However, the long-term, genomic effects of testosterone and its deficiency on the gene expression of Kv4 channels and their regulatory molecules remain unclear. The acute, non-genomic effects of testosterone and its derivatives on relaxation in VDSMs have been reported [25]. We initially examined the molecular components of the Kv4α and Kv4β subunits expressed in rat VDSMs using quantitative real-time PCR. We previously demonstrated that the longer variant of Kv4.3, Kv4.3L was expressed in rat VDSMs [19]. The basic properties of A-type K^+^ channels in rat VDSMCs have already been reported [26]; however, the molecular components of Kv4β subunits remain unknown. The present study showed that KChIP3, NCS1, and DPP6-S were abundantly expressed in rat VDSMs (Figure 2). However, the precise subunit stoichiometry of Kv4β subunits in rat VDSMCs remains unclear.

When I_A_ in VDSMCs was measured, 0.05 mM Cd^2+^ was added to a physiological external solution in order to suppress other ion channel currents (see Section 4.4.). Cd^2+^ has been shown to significantly affect the amplitude and some kinetics of I_A_ in a number of cell types [27,28]. Cd^2+^ induced a large rightward shift in the steady-state inactivation of cloned Kv4.3 currents [29]. Under control conditions, the half steady-state inactivation voltage (V_1/2_) of rat Kv4.3L was −51.4 ± 2.1 mV (*n* = 4) and the slope factor (S) was −6.3 ± 0.2. Following the addition of 0.05 mM Cd^2+^, the steady-state inactivation curve of Kv4.3L shifted to a positive direction (V_1/2_ = −40.2 ± 1.9 mV, *n* = 4), without significantly altering the S (6.6 ± 0.1) (unpublished data). As shown in Figure 4B, the V_1/2_ of the steady-state inactivation curve in I_A_ in rat VDSMCs was 20 mV more negative than that in Kv4.3L currents under the same conditions (approximately −60 mV). Furthermore, the times to half-inactivation from the peak at +20 mV were approximately 16 and 42 ms, respectively, and I_A_ in VDSMCs was inactivated 2.5-fold faster than Kv4.3L. The time constant of recovery from inactivation was one-third of Kv4.3L (τ = approximately 60 and 180 ms at −80 mV, respectively). Thus, Kv4.3L constitutes I_A_ in rat VDSMCs; however, the current feature that reconstituted Kv4.3L alone markedly differed from I_A_ in rat VDSMCs [19,29].

In heterologous expression systems, the co-expression of Kv4β subunits (KChIP3, NCS1, or DPP6) with the Kv4α subunit increases current density. However, modifications of Kv4α properties markedly differ. KChIP3 and NCS1 slow the rate of inactivation of Kv4 currents, whereas DPP6-S markedly accelerates the inactivation of Kv4 channels co-expressed with KChIPs [11]. Furthermore, DPP6-S induces a markedly larger negative shift in the steady-state inactivation of Kv4 currents than does KChIP3, and accelerates the rate of recovery from the inactivation of Kv4 channels [10,11]. NCS1 does not affect the voltage dependence of inactivation or the rate of recovery from the inactivation of Kv4 channels [30]. These findings suggest that DPP6-S but not KChIP3 or NCS1 strongly contributes to the modification of Kv4.3L channel kinetics in rat VDSMCs. The transcriptional expression level of DPP6S was 3-fold higher than those of KChIP3 and NCS1 (Figure 5).

The results of the present study demonstrated that the expression of Kv4.3 and KChIP3 was down-regulated in the VDSMs of castrated rats without changes in NCS1 or DPP6 expression (Figure 5 and Figure 6). KChIP3 expression was markedly decreased (more than 80%) by castration. These changes were almost completely compensated for by the treatment with MET. Several studies have revealed the hormonal regulation of A-type K^+^ channel expression by estrogen. Song et al. (2001) reported that estrogen markedly reduced Kv4.3 expression and function prior to parturition in the rat myometrium, and suggested that the membrane trafficking of Kv4.3 is controlled by estrogen [18]. Beckett et al. (2006) have showed the up-regulated expression of KChIP1 without changes in Kv4.3 expression levels in the gastrointestinal SM of ovariectomized mice [31]. Suzuki and Takimoto (2004) found that estrogen caused a switch in A-type K^+^ channel genes from the Kv4.3-KChIP4 complex to the Kv4.2-KChIP2 complex in the rat myometrium, during pregnancy [17]. Therefore, we examined whether a similar switching of A-type K^+^ channel genes was elicited in the VDSMs of castrated rats; however, the castration-induced up-regulation of the Kv4α and Kv4β subunits was not observed. Nerbonne et al. (2008) reported that the degradation of KChIP3 corresponding to the down-regulation of Kv4.3 contributed to the marked decreases in KChIP3 [32]. Therefore, another possibility to explain the marked decrease in KChIP3 by androgen deprivation is that testosterone might be responsible for maintaining the protein stability of the Kv4-KChIP complex.

The expression of Kv4.3L and KChIP3 was up-regulated during development in rat VDSMs (Appendix A). Turner et al. (2003) suggested that several growth factors, EGF, IGF, and FGF are involved in the testosterone-mediated development of the Wolffian duct [2]. Testosterone also activates mitogen-activated protein kinase (MAPK) and the cAMP response element binding (CREB) protein transcription factor in Sertoli cells [33]. Ca^2+^-dependent interactions between KChIP3 and CREB have been shown to represent a point of cross-talk between cAMP and Ca^2+^ signaling pathways in the nucleus [34]. These findings suggest that the gene expression of Kv4.3 or KChIP3 is associated with the signaling pathway mediating some growth factors, MAPK, or CREB protein transcription factors in VDSMs.

As shown in Figure 6D and Figure 7C, in the rat VD, DPP6-S was glycosylated by castration, and deglycosylated by the MET-treatment. Iusem et al. (1984) reported that glucosyl and mannosyl transferase activities decreased after castration in rat epididymal microsomes, and the depleted mannosyl transferase activity was restored to control values by the administration of testosterone [35]. This finding suggests that the glycosylation of DPP6-S is regulated by testosterone in VDSMs. On the other hand, Kv4 channels contain no consensus sequence for glycosylation on their extracellular surface. However, in ventricular myocytes, the removal of sialic acid, a negatively charged sugar residue was found to reduce the number of K^+^ channels at the plasma surface, resulting in a reduction in I_to_ produced by Kv4 channels [36]. Although it currently remains unclear whether the glycosylation of DPP6-S by castration affects Kv4 channel kinetics, the present results suggest that the glycosylation of DPP6-S is responsible for Kv4.3 channel protein turnover, and, thus, might compensate for the decrease in I_A_ by castration.

## 4. Materials and Methods

### 4.1. Surgery and Hormonal Manipulations

Male Wistar rats were castrated at 3 weeks of age under ether anesthesia. Castrated rats were divided into three groups—(i) not injected (castrated group), (ii) injected daily p.o. with 100 mg/kg 17α-methyltestosterone (MET) in sesame oil (MET-treated group), and (iii) injected daily p.o. with vehicle alone (vehicle-treated group). After 4 weeks, rats in each group were anesthetized and killed by bleeding, and the body weights and the wet weights of various tissues (VD, prostate, seminal vesicles, heart, liver, and kidney) were measured. Systolic blood pressure was simultaneously measured in unanesthetized and restrained rats warmed to 39 °C in the morning (9–11 AM) using the tail-cuff method. All experiments were performed in accordance with the guiding principles for the care and use of laboratory animals (the Science and International Affairs Bureau of the Japanese Ministry of Education, Science, Sports and Culture) and also with the approval of the Ethics Committee of Nagoya City University.

### 4.2. Total RNA Extraction, Reverse Transcription, and Real-Time PCR

Total RNA was extracted from rat tissues and reverse-transcribed as previously reported [37]. The PCR amplification profile was as follows—at 95 °C for 15 s and at 60 °C for 60 s according to the protocol recommended by Applied Biosystems (Foster City, CA, USA). ACTB primers were used to confirm that the products generated were representative of RNA. Each amplified product was sequenced by the chain termination method with an ABI PRIZM 3100 genetic analyzer (Applied Biosystems). Real-time quantitative PCR was performed using Syber Green chemistry on an ABI 7000 sequence detector (Applied Biosystems), as reported previously [38]. Unknown quantities relative to the standard curve for a particular set of primers were calculated, yielding the transcriptional quantitation of gene products relative to the endogenous standard, ACTB. The following PCR primers were used—Kv4.2 (GenBank accession no. NM_031730, 1364–1464), amplicon = 101 bp; Kv4.3 (NM_031739, 324–425), 102 bp; KChIP1 (NM_022929, 141–241), 101 bp; KChIP2 (NM_020095, 283–384), 102 bp; KChIP3 (AF297118, 124–262), 139 bp; KChIP4 (NM_181365, 623–728), 107 bp; KChAP (AF032872, 531–640), 110 bp; NCS1 (NM_024366, 415–515), 101 bp; DPP6 (M76426, 991–1093), 103 bp; DPP6–L (M76426, 118–228), 111 bp; DPP6–S (M76427, 18–134), 117 bp; DPP10 (AY557199, 744–846), 103 bp; and ACTB (NM_031144, 338–438), 101 bp. In Figure 4, the following PCR primers that were specific for rat DPP6–S were designed—(M76427, −3–321), 324 bp; (285–703), 419 bp; (537–815), 479 bp; (792–1236), 437 bp; (1065–1529), 465 bp; (1353–1852), 500 bp; (1507–2019), 513 bp; and (1772–2421), 650 bp.

### 4.3. Western Blotting

Membrane fractions of the rat VD were prepared according to the Alomone Labs protocol (Jerusalem, Israel), and the protein contents were measured using a protein assay kit (Bio-Rad Laboratories, Hercules, CA, USA) with bovine serum albumin as a standard. Protein samples were subjected to SDS-PAGE on a 10% polyacrylamide gel and the proteins were then transferred to polyvinylidene difluoride (PVDF) membranes (Bio-Rad Laboratories). Membranes were blocked with 0.1% Tween 20 in PBS. Blots were incubated with affinity purified polyclonal antibodies specific for Kv4.3 (Alomone Labs), KChIP3 (Santa Cruz Biotechnology, Santa Cruz, CA, USA), and DPP6 (gifted by Dr. Yoshio Misumi, Fukuoka University, Japan) overnight, and then incubated with anti-rabbit or anti-goat horseradish peroxidase-conjugated IgG for 1 h. Regarding deglycosylation, resuspended membrane fractions (50 μg) were treated with the Enzymatic Protein Deglycosylation kit (Sigma-Aldrich, St. Louis, MO, USA) according to the user’s manual. An enhanced chemiluminescence detection system (Amersham Biosciences, Piscataway, NJ, USA) was employed for the detection of the bound antibody. The resulting images were analyzed by a LAS-1000 (Fuji Film, Tokyo, Japan) and the digitized signals were quantified with Image Gauge software (Version 3.3, Fuji Film). In summarized results, relative protein expression levels in the control were expressed as 1.0.

### 4.4. Cell Isolation and Electrophysiology

Single SMCs were enzymatically isolated from VD of male Wistar rats (7 week-old) using previously reported method [7,27]. As previously reported, we confirmed that VDSMCs were morphologically elongated and induced contraction and intracellular Ca^2+^ elevation by application of 100 µM norepinephrine, as reported previously [39,40]. Whole-cell voltage clamping was applied to single cells with patch pipettes, using a CEZ-2400 amplifier (Nihon Kohden, Tokyo, Japan). When the whole-cell currents of I_A_ on VDSMCs were recorded, an external medium contained the following (in mM)—NaCl 107, tetraethylammonium (TEA) chloride 30, KCl 5.9, CaCl_2_ 2.2, MgCl_2_ 1.2, CdCl_2_ 0.05, glucose 14, and HEPES 10 (pH 7.4). The pipette solution contained (mM); KCl 140, MgCl_2_ 4, ATP-2Na 5, EGTA 5 and HEPES 10 (pH 7.2).

The voltage-dependence of activation and steady-state inactivation and the rate of recovery from inactivation were measured using a conventional double-pulse protocol as described previously [12,13]. The membrane potential was held at −80 mV and depolarized to test potentials for 3 ms in order to activate I_A_ and then to −50 mV to measure the tail current. The tail current amplitude was normalized to the maximum in each cell and plotted against the test potentials. Data were fit with the Boltzmann function and the voltages required for half maximal activation and S were then assessed. This double pulse was sequentially applied once every 10 s. The voltage-dependence of I_A_ inactivation was also evaluated using a double-pulse protocol. I_A_ was activated and inactivated by depolarization from −80 mV to test potentials for 1 s, and the remaining available channels were then activated by depolarization to +20 mV. Normalization and the fitting of data to the Boltzmann function were performed. This double pulse was sequentially applied every 10 s. All experiments were performed at room temperature (23 ± 1 °C). Data acquisition and analyses were conducted using software (AQ and cell soft) developed in laboratory of Dr. Giles (University of Calgary, Calgary, AB, Canada).

### 4.5. Statistical Analysis

The significance of differences among multiple groups was evaluated by Scheffé’s test, after conducting an ANOVA. Results with a *p*-value of less than 0.05 or 0.01 were considered to be significant. Data are presented as means ± SEM.

## 5. Conclusions

The results of the present study suggest that elevated levels of testosterone are a prerequisite for the up-regulation of the A-type K^+^ channel components, Kv4.3L and KChIP3 in rat VDSMs. The molecular mechanisms underlying the regulation of Kv4.3 and KChIP3 expression by testosterone remain unclear; however, testosterone might contribute to the membrane trafficking and protein degradation of the Kv4.3-KChIP3 complex. It is also important to note that DPP6-S might mainly contribute to modifications to Kv4.3L inactivation properties in rat VDSMs. Furthermore, the present results suggest that the increase in testosterone levels in growth processes controls A-type K^+^ channel expression in VDSMs. Recent studies showed testosterone-induced up-regulation of delayed rectifier Kv7.1 K^+^ channel via a transcription factor SP1 [41] and Ca^2+^-activated K^+^ channel K_Ca_3.1 and K_Ca_2.3 via endothelial nitric oxide synthase (eNOS) signaling [42]. Further investigations on the mechanisms underlying the testosterone-mediated regulation of K^+^ channels will provide important information for the reproductive dysfunction induced by testosterone therapy for aging men and androgen deprivation therapy for patients with prostate cancer.

## Figures and Tables

**Figure 1 ijms-20-04073-f001:**
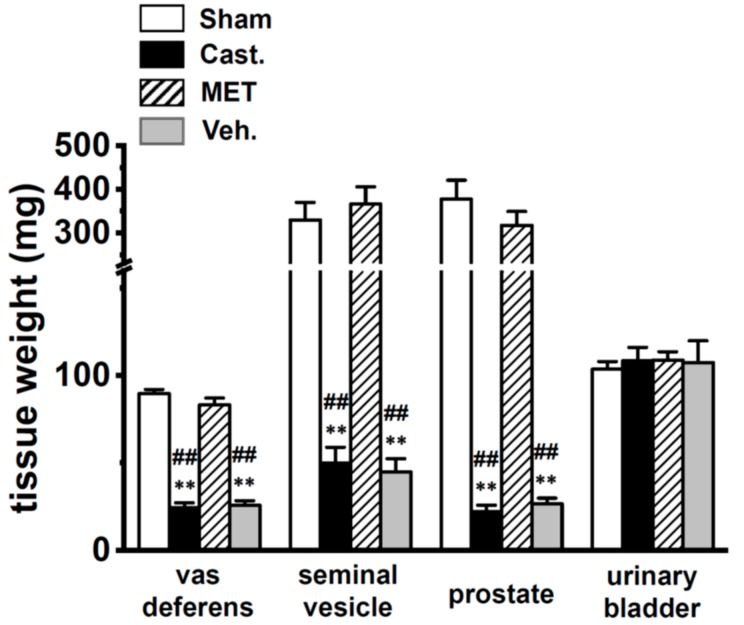
Changes in tissue weights by castration. Weights of the vas deferens, seminal vesicles, prostate, and urinary bladder in each group of male rats. Open columns—sham-operated group (**Sham**), black columns—castrated group (**Cast.**), hatched columns—methyltestosterone (MET)-treated group (**MET**), and gray columns—vehicle-treated group (**Veh.**). ** *p* < 0.01 versus sham-operated; ^##^
*p* < 0.01 versus MET-treated (*n* = 9).

**Figure 2 ijms-20-04073-f002:**
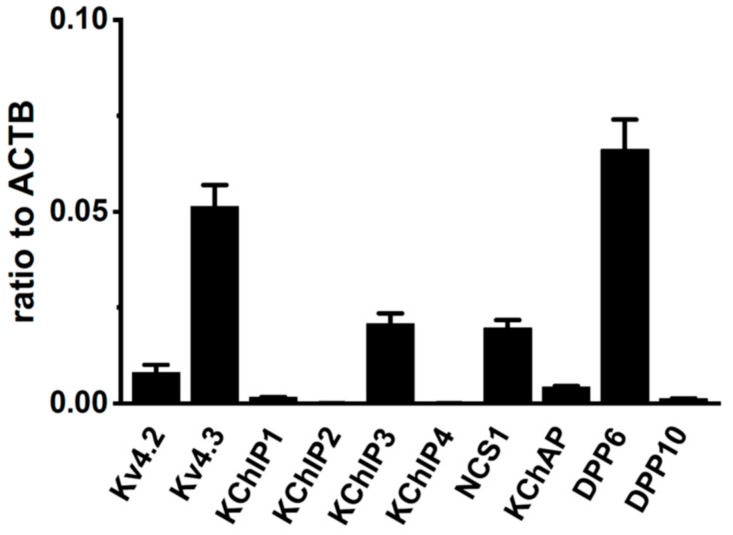
Molecular identification of A-type K^+^ channel α and β subunits (Kv4α and Kv4β) in vas deferens smooth muscles (VDSMs). Quantitative real-time PCR for Kv4α (Kv4.2, Kv4.3) and Kv4β (KChIP1-3, KChAP, DPP6, and DPP10) expression in the vas deferens of 7 week-old male rats (*n* = 6). Values are shown for steady-state transcripts relative to β-actin (ACTB) in the same preparation.

**Figure 3 ijms-20-04073-f003:**
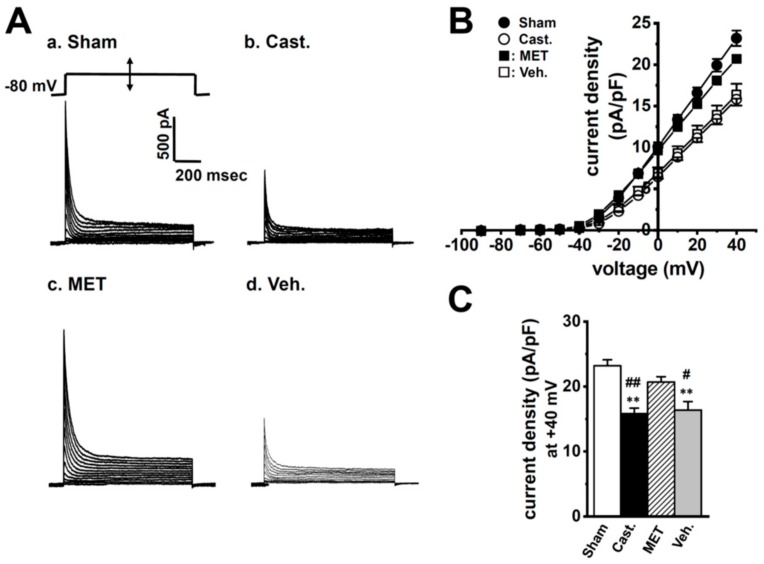
Effects of castration on transient outward currents, I_A_ density in rat VDSMCs. (**A**) I_A_ was elicited by a 1000 ms depolarizing-voltage-step between −70 and +40 mV from a holding potential (−80 mV) with 10 mV increments once every 10 s in the VDSMCs of each group. (**B**) Current density–voltage relationship in each group. (●) sham-operated (**Sham**) (*n* = 15), (○) castrated (**Cast.**) (*n* = 20), (■) MET-treated (**MET**) (*n* = 17), and (□) vehicle-treated (**Veh.**) (*n* = 16). (**C**) Summarized data of current density at +40 mV in four groups. ** *p* < 0.01 versus Sham; ^#^, ^##^: *p* < 0.05, 0.01 versus MET.

**Figure 4 ijms-20-04073-f004:**
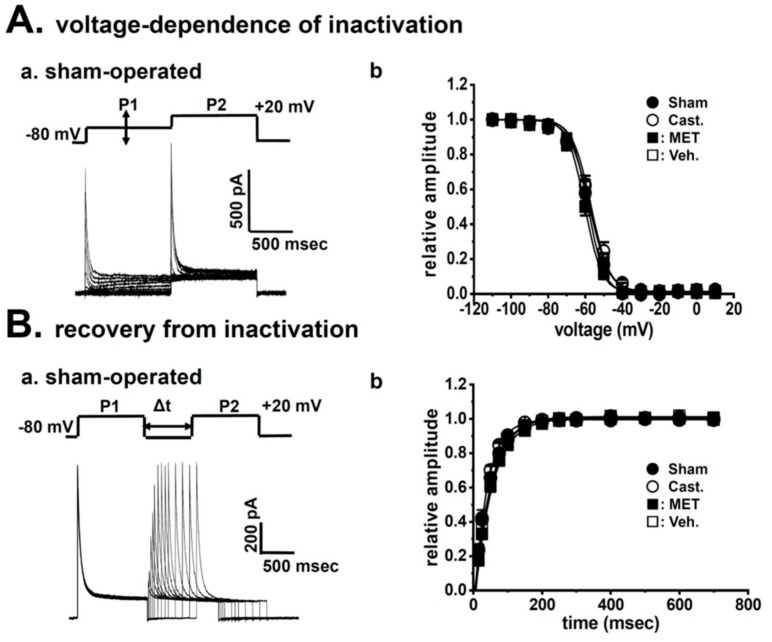
Effects of castration on I_A_ kinetics in rat VDSMCs. (**Aa**) The steady-state voltage dependence of inactivation was assessed by the conventional double-pulse protocol. I_A_ was activated and inactivated with 1 s step depolarization from −80 mV to the test potentials (**P1**), and the remaining channels were then activated by a second depolarization to +20 mV (**P2**). (**Ab**) Normalization of current records and fitting of data to the Boltzmann equation were performed, and the voltages required for the half-maximal inactivation and slope factors were obtained from the best fitting relationship. (●) Sham-operated (**Sham**) (*n* = 14), (○) castrated (**Cast.**) (*n* = 16), (■) MET-treated (**MET**) (*n* = 14), and (□) vehicle-treated (**Veh.**) (*n* = 14). (**Ba**) The paired-pulse protocol was applied to assess the time course of recovery from the inactivation of I_A_ at −80 mV. (**Bb**) Summarized data obtained from eight cells were plotted as the relative amplitude of I_A_ (I_P2_/I_P1_) against Δt (ms). The recovery time course was described by a single exponential function. (●) Sham (*n* = 10), (○) Cast. (*n* = 15), (■) MET (*n* = 15), and (□) Veh. (*n* = 14).

**Figure 5 ijms-20-04073-f005:**
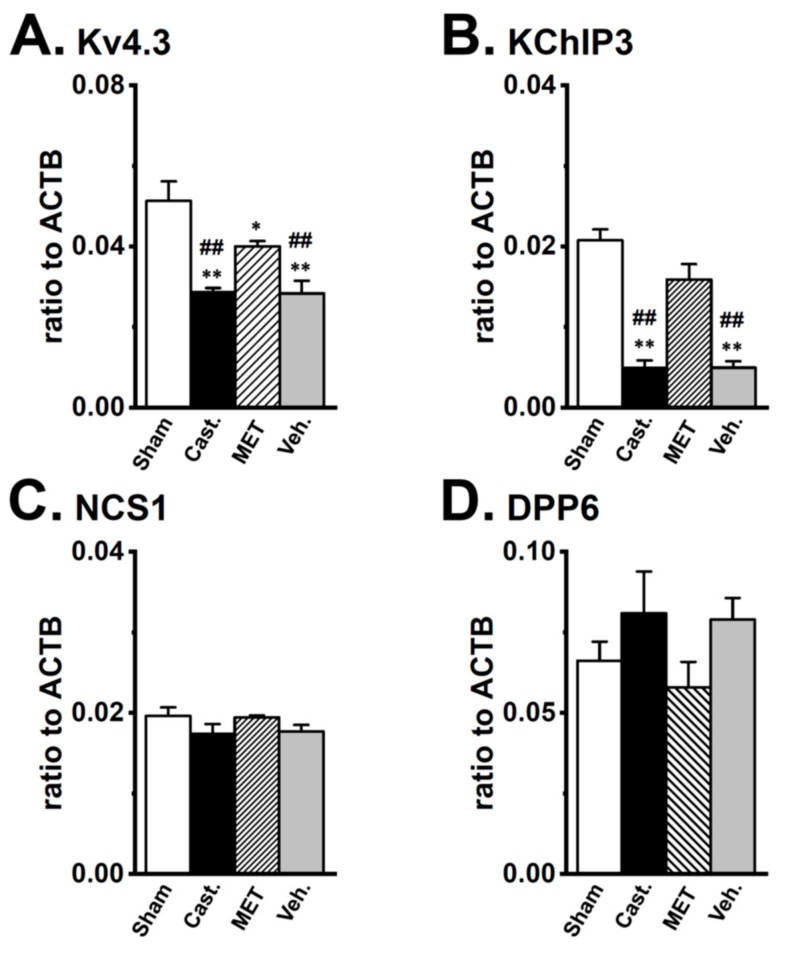
Changes in the transcriptional expression of Kv4α and Kv4β by castration in rat VDSMs. The expression of Kv4.3, KChIP3, NCS1, and DPP6 transcripts in each group was examined by real-time PCR analysis. Values are shown for steady-state transcripts relative to ACTB in the same preparation. (**A**) Kv4.3, (**B**) KChIP3, (**C**) NCS1, (**D**) DPP6 (*n* = 6 for each). Open columns—sham-operated group (**Sham**), black columns—castrated group (**Cast.**), hatched columns—MET-treated group (**MET**), and gray columns—vehicle-treated group (**Veh.**). *, ** *p* < 0.05, 0.01 versus Sham; ^##^
*p* < 0.01 versus MET.

**Figure 6 ijms-20-04073-f006:**
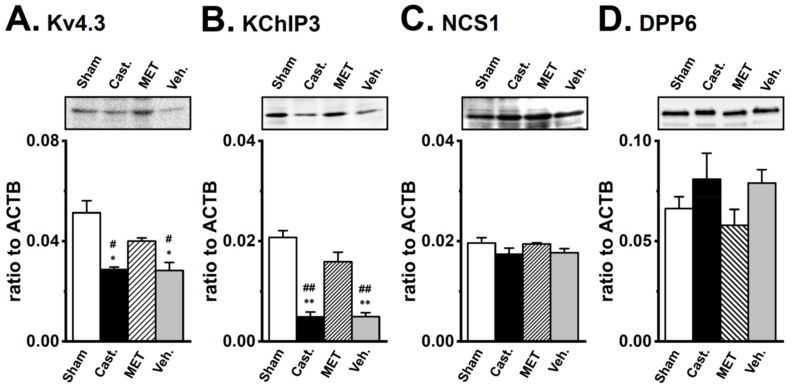
Changes in the protein expression of Kv4α and Kv4β by castration in rat VDSMs. The expression of the Kv4.3, KChIP3, NCS1, and DPP6 proteins in vas deferens of each group was examined by Western blotting. The membrane fraction extracted from the VDSMs of each group was immunoblotted with the anti-Kv4.3 (1:200), anti-KChIP3 (1:200), anti-NCS1 (1:200), and anti-DPP6 (1:500) antibodies. Values were expressed as a ratio to those in the VDSMs of sham-operated rats. (**A**) Kv4.3, (**B)** KChIP3, (**C**) NCS1, and (**D**) DPP6 (*n* = 4 for each). Open columns—Sham, black columns—Cast., hatched columns—MET, and gray columns—Veh. *, ** *p* < 0.05, 0.01 versus sham-operated; ^#^, ^##^
*p* < 0.05, 0.01 versus MET-treated.

**Figure 7 ijms-20-04073-f007:**
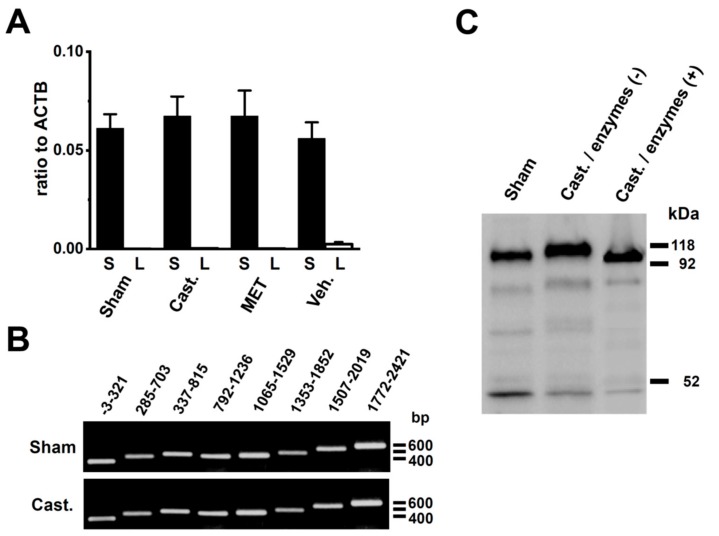
Identification of DPP6 isoforms expressed in rat VDSMs. (**A**) Quantitative real-time PCR analysis of the transcripts of the DPP6 isoforms, DPP6-L and DPP6-S in the VDSMs of each group by real-time PCR. Values are shown for steady-state transcripts relative to ACTB in the same preparation. Sham-operated (**Sham**); castrated (**Cast.**); MET-treated (**MET**); and vehicle-treated (**Veh.**) (*n* = 6 for each). (**B**) Expression of DPP6-S isoforms in the VDSMs of each group (sham-operated and castrated) by RT-PCR. Each PCR product of DPP6-S was identified by ethidium bromide staining: −3–321 (324 bp), 285–703 (419 bp), 537–815 (479 bp), 792–1236 (437 bp), 1065–1,529 (465 bp), 1353–1852 (500 bp), 1507–2019 (513 bp), and 1772–2421 (650 bp). (**C**) Glycosylation of DPP6-S by castration. The VDSM membrane fractions of sham-operated and castrated rats were prepared as described in Section 4.3. Before (−) and after (+) deglycosylation, 50 µg of protein/lane was loaded. An anti-DPP6 antibody was used to probe for glycosylated and deglycosylated DPP6.

**Table 1 ijms-20-04073-t001:** Electrophysiological parameters in rat VDSMCs of each group—sham-operated (**Sham**), castration (**Cast.**), MET-treated (**MET**), and vehicle-treated (**Veh.**). *, **: *p* < 0.05, 0.01 versus Sham; ^#^, ^##^: *p* < 0.05, 0.01 versus MET.

	Sham	Cast.	MET	Veh.
**current amplitude at +40 mV**				
current (nA)	1.22 ± 0.06	0.63 ± 0.04 **^,##^	1.11 ± 0.07	0.56 ± 0.05 **^,##^
capacitance (pF)	53.2 ± 2.5	39.8 ± 0.9 **^,##^	53.2 ± 2.3	34.8 ± 1.4 **^,##^
current density (pA/pF)	23.2 ± 0.9	15.8 ± 0.9 **^,##^	20.7 ± 0.8	16.4 ± 1.3 **^,#^
	*n* = 15	*n* = 20	*n* = 17	*n* = 16
**activation/inactivation rate at +20 mV**				
time to peak (ms)	4.35 ± 0.25	4.13 ± 0.23	4.65 ± 0.25	4.44 ± 0.26
inactivation rate, t_1/2_ (ms)	18.0 ± 1.8	14.6 ± 0.6	19.7 ± 1.4	19.3 ± 1.8
	*n* = 15	*n* = 20	*n* = 17	*n* = 16
**voltage-dependence of activation**				
V_1/2_ (mV)	−13.9 ± 2.4	−9.2 ± 3.5	−10.4 ± 6.1	−11.3 ± 4.0
slope factor	12.7 ± 0.7	9.0 ± 1.0	11.7 ± 1.0	13.5 ± 1.8
	*n* = 5	*n* = 6	*n* = 5	*n* = 5
**voltage-dependence of inactivation**				
V_1/2_ (mV)	−58.6 ± 1.4	−57.0 ± 1.5	−60.3 ± 1.1	−57.7 ± 0.9
slope factor	4.47 ± 0.21	4.97 ± 0.15	4.58 ± 0.10	5.36 ± 0.23 *^,#^
	*n* = 14	*n* = 16	*n* = 14	*n* = 14
**recovery from inactivation**				
τ (ms)	47.9 ± 8.9	37.5 ± 3.1	51.6 ± 5.2	50.2 ± 5.2
	*n* = 10	*n* = 15	*n* = 15	*n* = 14

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
