# Peer review of "Castration Induces Down-Regulation of A-Type K^+^ Channel in Rat Vas Deferens Smooth Muscle"

_ijms, 2019, doi:10.3390/ijms20174073_

Round 1

Reviewer 1 Report

The authors identify the molecular components of Kv4 subunits in the rat VDSMs. They further identify the influence of testosterone on the expression levels of Kv4 alpha and beta subunits in rat VD by comparing the levels after castration. A significant rescue of the changes in expression levels after methyl-testosterone treatment further confirms the testosterone-mediated effects underlying such changes. Overall their findings suggest that changes in testosterone levels during growth or with disorders of testosterone production/functions could control A-type K+ channel expression in VDSMs. Molecular mechanisms such as DPP6 involved in the regulation of Kv4 subunits are revealed. The study is well designed and the data are interpreted appropriately. The authors need to revise the manuscript to improve the readability.

Author Response

We really appreciate to the reviewer’s careful reading and adequate suggestion. This manuscript is carefully elaborated over again by all authors.

Reviewer 2 Report

Ohya S et.al., mainly studied to investigate Castration induces down-regulation of A-type K+ channel and these effects were reversed by methyltestosterone in rat vas deferens smooth muscle. Authors have delineated this by demonstrating an invitro and physiological approach to show that Kv4 channel mediated currents are important for VD relaxation. Although this is an interesting topic, but more data should be provided to confirm their hypothesis.       

However, some major concerns to be addressed before accepting for publication.

1.    What is the rationale for choosing MET drug and it was not mentioned anywhere in the introduction, and suddenly appeared in the first figure?

2.    SMCs isolation: Characterization was not provided, and which passage cells were employed to test currents and molecular analysis?

3.    How does the authors claim that these currents are mediated through Kv channels in VDSMcs. Did they show with any specific activator or inhibitor?

4.    Gene expression pattern of Kv channels in castrated and sham operated animals is important and should be provided. It is interesting to note that DPP6 (Kv4β channel) is comparable to Kv4α (Kv4.3) but did not change after castration?

5.    Western blot: loading controls are not provided and it is interesting to note that compare to sham MET treated Kv4.3 expression is high in the blot, but quantification shows less compared to sham? Authors should reconcile this.

Further authors claim that its membrane fraction? It should also be shown with the whole cell/tissue lysate?

6.    7th figure is inserted twice. The resolving 5kDa difference (due to post translational modification) in 10% gel is not convincing.

Author Response

We really appreciate to the reviewer’s valuable comments and suggestions. Changes in the text are shown by red characters with underline.

1. 17α-methyltestosterone (MET) is a synthetic derivative of testosterone given by the oral route without loss of bioavailability. In accordance with the reviewer’s suggestion, we added the description in ‘Abstract’ and ‘Introduction’ sections. (Page 1, line 27 and Page 2, line 20-line 21.

2. In this study, only freshly-isolated vas deferens smooth muscle cells were used. As reported in our previous studies, an isolated vas deferens smooth muscle cell is morphologically elongated (Ohya et al., 2001) and shows contraction and intracellular Ca2+ elevation by application of 100 µM norepinephrine (Ohno et al., 2009). We added the description on this in ‘Materials and Methods’ section (Page 11, line 30-line 32).

Ohya, S.; Yamamura, H.; Muraki, K.; Watanabe, M.; Imaizumi, Y. Comparative study of the molecular and functional expression of L-type Ca2+ channels and large-conductance, Ca2+-activated K+ channels in rabbit aorta and vas deferens smooth muscle. Pflugers Arch. 2001, 441, 611-620. doi: 10.1007/s004240000463

Ohno, A.; Ohya, S.; Yamamura, H.; Imaizumi, Y. Regulation of ryanodine receptor-mediated Ca2+ release in vas deferens smooth muscle cells. J. Pharmacol. Sci. 2009, 110, 78-86. doi: 10.1254/jphs.09037FP

3. As described in Section 4.4., the external medium includes 1.2 mM CdCl2 and 30 mM TEA to suppress L-type, voltage-gated Ca2+ currents, voltage-gated K+ currents, and large-conductance Ca2+-activated K+ Also, the pipette solution includes 5 mM EGTA to suppress large-conductance Ca2+-activated K+ currents. Under this condition, A-type K+ current component was observed and was disappeared by application of 1 mM 4-aminopyridine, a specific Kv4 channel blocker. We added these descriptions in ‘Results’ section (Page 2, line 44-line 48). Thank you for your adequate indication.

4. Our previous study showed that rat VDSM expressed Ca2+-activated KCa1 and delayed rectifier Kv2.1 K+ channels (Ohya et al., 2001). We recently found that castration decreased KCa1.1 K+ currents. Castration promoted protein degradation of KCa1.1 without changing transcriptional level of it. In contrast, Kv3.1 activity and expression were increased by castration. Currently, we are investigating the molecular mechanisms of castration-induced promotion of KCa1.1 protein degradation and compensatory Kv3.1 up-regulation. Therefore, we would like to focus on castration-induced down-regulation of A-type K+ channel components but not delayed rectifier/Ca2+-activated K+ channels in this manuscript.

Ohya, S.; Tanaka, M.; Watanabe, M.; Imaizumi Y. Diverse expression of delayed rectifier K+ channel subtype transcripts in several types of smooth muscles of the rat. J. Smooth Muscle Res. 2000, 36, 1010-115.

5. In our preliminary experiments, administration of MET for 4 weeks in normal rats did not affect the expression levels of Kv4.3 transcripts and IA density (n=2 to 3). Therefore, we compared to sham-operated group in this study. We showed the developmental changes in the gene and protein expression levels of Kv4.3, KChIP3, NSC-1, and DPP6 (Figure S1). Although not shown in the figure, the expression levels of them in 11 week-old rats were almost similar to those in 7 week-old ones, suggesting that A-type K+ channel components may be fully expressed in VDSMs of 7 week-old male rats. We briefly described the preliminary results in VDSMs of MET-administrated normal male rats (Page 3, line 4-line 5).

Nerbonne and Kass (Physiol. Rev., 2005) showed that interaction of Kv4 with KChIP and DPP in their review article (Physiol. Rev., 2005). Also, our collaborator, Prof. Choe (Salk institute, USA) showed the direct interaction of Kv4 with KChIP (Zhou et al., Neuron, 2004). NCS-1 is translocated to plasma membrane (Ames & Lim, 2012). This is the reason why we used membrane fraction proteins in this study.

Nerbonne, J.M.; Kass, R.S. Molecular physiology of cardiac repolarization. Physiological. Rev. 2005, 85, 1205-1253.

Zhou, W.; Qian, Y.; Kunjilwar, K.; Pfaffinger, P.J.; Choe, S. Structural insights into the functional interaction of KChIP1 with Shal-type K+ channels. Neuron. 2004, 41, 573-586.

Ames, J.B.; Lim, S. Molecular structure and target recognition of neuronal calcium sensor proteins. Biochim. Biophys. Acta. 2012, 1820, 1205-1213.

6. We cannot find repetitive insertion of Figure 7 in our submitted file. We will confirm this problem to the editorial office.

Next, we agree with the reviewer’s comment that the resolving 5 kDa difference (due to post translational modification) in 10% gel is not convincing (Fig. 7). However, it takes over 1 month to obtain new data, and there are no stock of Enzymatic Protein Deglycosylation kit (Sigma-Aldrich) in Japan. Finally, we removed the description ‘approxilmately 5 kDa’ (Page 4, line 31 in original manuscript) because this value is not deterministic in our gel data.